

# Alpine tectonic wedging and crustal delamination in the Cantabrian Mountains (NW Spain)

Jorge Gallastegui[1], Javier A. Pulgar[1], Josep Gallart[2]

[1]Departamento de Geología, Universidad de Oviedo, Jesús Arias de Velasco s/n, 33005 Oviedo, Spain
[2]Instituto Ciencias Tierra Jaume Almera-CSIC, Lluís Solé Sabarís s/n, 08028 Barcelona, Spain

*Correspondence to*: J. Gallastegui (jorge@geol.uniovi.es)

**Abstract.** The Cantabrian Mountains have been interpreted as a Paleozoic basement block uplifted during an Alpine deformation event that led to the partial closure of the Bay of Biscay and the building of the Pyrenean range in the Cenozoic. A detailed interpretation of deep seismic reflection profile ESCIN-2 and the two-dimensional seismic modelling of the data

allowed us to construct a N-S geological cross-section along the southern border of the Cantabrian Mountains and the transition to the Duero Cenozoic foreland basin, highlighting the Alpine structure. The proposed geological cross-section has been constrained by all geophysical data available, including a 2-D gravity model constructed for this study as well as refraction and MT models from previous studies. A set of thrusts vergent to the S, dipping 30° to 36°, cut the upper crust and are responsible for the uplift and the main Alpine deformation in the Cantabrian Mountains. A conspicuous reflection Moho

shows that the crust thickens northwards from the Duero basin, where subhorizontal Moho is 32 km deep, to 47 km in the northernmost end of ESCIN-2 where it is dipping to the north beneath the Cantabrian Mountains. Further north, out of the profile, Moho reaches a maximum depth of 55 km according to wide-angle/refraction data. ESCIN-2 indicates the presence of a tectonic wedge of the crust of the Cantabrian margin beneath the Cantabrian Mountains, which is indented from north to south into the delaminated Iberian crust, forcing its northwards subduction.

**1 Introduction**

The Cantabrian Mountains constitute an E-W oriented range that extends more than 250 km, and is the western extension of the Pyrenean range (Fig. 1). The tectonic history of the area is long and complex, spanning from lower Paleozoic to upper Miocene times. The eastern end of the range has been interpreted as a Paleozoic basement block, which is part of the Variscan chain of Western Europe, that was later uplifted and slightly deformed during the Alpine cycle (Alonso et al.,

1996). The studied area is located in the outermost section of the Iberian Variscan foreland thrust and fold belt, where Variscan deformation took place under shallow conditions with scarce metamorphism and cleavage (Matte, 1991, Pérez-Estaún and Bastida, 1991). The well known Variscan structure of the mountain range has been studied for a long time (e.g. Julivert, 1971; Pérez-Estaún et al., 1988; Dallmeyer and Martínez-García, 1990; Alonso et al. 2009), and more recent studies have focused on the Alpine structure (Alonso et al., 1996; Pulgar et al., 1996, 1999; Gallastegui, 2000; Gallastegui et al.,



2002, Martín-González et al., 2011; Pedreira et al., 2015). The Alpine history is directly related to the post-Variscan evolution of the North Iberian Margin that started by a Permo-Triassic rifting stage that was followed by an approximately N-S extensional episode, triggered by the opening of the Atlantic Ocean and the Bay of Biscay during Late Jurassic and Early Cretaceous times. The basins that developed in the passive margin remained stable during the sea-floor spreading

phase that took place in the Bay of Biscay in the upper Cretaceous (Martínez-García, 1982; Vegas and Banda, 1982; Boillot et al., 1984; Boillot y Malod 1988; Verhoef and Srivastava, 1989). The tectonic setting changed to NW-SE compression as a result of the convergence between plates in Cenozoic times after the change to the N in the motion of the African plate. The new compressional setting resulted in: i) the building of the Pyrenees further to the East as a result of a continental collision, ii) the partial closure of the Bay of Biscay, iii) the tectonic inversion of the North Iberian Margin that evolved to an active

margin during the Cenozoic and iv) the uplift of the Cantabrian Mountains. Although the structure observed in the studied area is mainly Variscan, some Alpine structures are superposed. These later structures include i) favourably oriented reworked Variscan thrusts and Mesozoic normal faults and ii) newly formed faults (Alonso et al., 1996, Pulgar et al., 1999). Evidences of the Alpine deformation are also present further to the N in the North Iberian continental platform and abyssal plain, where north verging thrusts developed, previous extensional Mesozoic faults were inverted and the sedimentary cover

was folded (Álvarez-Marrón et al. 1997; Gallastegui, 2000; Gallastegui et al., 2002; Fernández Viejo et al., 2011, 2013). The present relief of the area is Alpine and its erosion supplied the detritus that filled the main synorogenic basins that developed in its boundaries: the inland Duero foreland basin in the South and the offshore Iberian Platform and abyssal plain basins in the North.

The ESCIN and MARCONI programs and other related projects were planned to study the deep crustal structure and

evolution of the Cantabrian Mountains and continental margin from deep seismic reflection and refraction/wide-angle data (Pérez-Estaún et al., 1994; Pulgar et al., 1995, 1996; Álvarez-Marrón et al., 1996, 1997; Gallastegui et al., 1997; Ayarza et al., 1998; Fernández-Viejo et al., 1998, 2000; Gallastegui, 2000; Gallastegui et al., 2002; Fernández-Viejo and Gallastegui 2005; Fernández-Viejo et al., 2011, 2012). These studies showed that the deformation of the area due to the Alpine compression not only affected the shallow crustal levels, but also the deeper ones. An important Alpine age crustal root

developed under the uplifted Cantabrian Mountains and has been imaged in deep seismic reflection profile ESCIN-2 (Pulgar et al., 1995) and in several refraction/wide-angle experiments (Pulgar et al., 1996; Fernández-Viejo et al., 1998, 2000). This thickening of the crust has also been interpreted in magnetotelluric profiles (Pous et al, 2001) and explored using teleseismic receiver function analysis of P to S conversions at main crustal interfaces (Díaz et al., 2003, 2009). Pedreira et al. (2003, 2007) and Díaz et al. (2012) proved that this structure extends eastwards beneath the Basque-Cantabrian basin and is the

continuation of the same crustal root interpreted beneath the Pyrenees (ECORS Pyrenees Team, 1988; Bois et al., 1990). Crustal depth models compiled from deep sounding experiments (Gallastegui, 2000;) also show this crustal thickening with Moho depths up to 50 km in the NW of the Iberian Peninsula (Fig. 2) and the continuity of this E-W crustal structure from the Pyrenees to the Cantabrian Mountains (Díaz and Gallart, 2009). This paper focuses on the N-S deep crustal structure of the southern part of the Cantabrian Mountains and the transition to the adjacent Cenozoic Duero basin imaged in deep





seismic reflection profile ESCIN-2. A geological cross-section of the area has been constructed integrating published data (ESCIN-2, a refraction profile and a conductivity model) with a depth model of ESCIN-2 obtained by 2-D seismic modelling, a gravity model coincident with ESCIN-2 and all geological/geophysical data available (wide-angle/refraction and MT data).

**2. Geology along ESCIN-2 deep seismic reflection profile**

Deep seismic reflection profile ESCIN-2 runs across two geological domains with unlike lithologies and structure, which differ in the age of the materials that crop out and their degree of deformation. The northern half of the seismic line traverses the deformed Paleozoic rocks of the Cantabrian Mountains. In the southern half these Paleozoic rocks constitute the basement of the mainly undeformed subhorizontal Cenozoic materials that fill the Duero basin. An outcrop of almost vertical
to overturned Upper Cretaceous and Cenozoic rocks separates both areas (Fig. 3).
In the northern domain the structure is mainly Variscan with minor Alpine structures superposed. The Variscan deformation affected a kilometric thick pile of clastic and carbonate rocks that range from Silurian to Carboniferous. The line runs across strike of approximately northwards emplaced Carboniferous thrusts and related folds. These structures are part of a thin-skinned foreland thrust and fold belt, the so-named Cantabrian Zone (Julivert, 1971; Pérez-Estaún et al., 1988; Pérez Estaún
et al., 1994; Alonso et al., 2009). It is difficult to infer the significance of the Mesozoic extensional deformation superposed in the area, due to the lack of Mesozoic outcrops. However, Alonso et al. (1996) deduced extensional faults in the area by projecting those that affect the Mesozoic rocks located to the east in the nearby Basque-Cantabrian basin. The Alpine compressional deformation in the area was mainly accommodated by the reactivation of previous structures: Variscan folds and thrusts and Mesozoic normal faults. The Ruesga-Ubierna fault (see R.U.F. in Fig. 3) is an example of a major Mesozoic
normal fault, which was later inverted in Alpine times (Espina et al., 1996). The development of important Alpine structures is restricted to the southern border of the Cantabrian Mountains, where the overall Alpine structure is a wide monoclinal fold. Alonso et al. (1996) and Pulgar et al. (1999) interpreted it as a major fault-bend fold related to a large N-dipping basement-thrust ramp. This buried basement thrust (main Alpine thrust) cuts across the upper crust and is responsible for the basement-cored uplift of the Cantabrian Mountains. The frontal limb of the fold makes up the boundary between the
Cantabrian Mountains and the Cenozoic Duero basin where Mesozoic-Cenozoic rocks are overturned. The progressive rotation of the limb during the Alpine deformation was recorded by a syntectonic unconformity that developed in the Cenozoic sediments along the northern margin of the Duero basin (García-Ramos et al., 1982; Alonso et al., 1996). This basin is considered the foreland basin of the uplifted Cantabrian Mountains.
The surface structure of the Duero basin in the southern domain is simpler. The overturned Mesozoic rocks that constitute
the border of the basin, progressively become horizontal towards the S and make the base of the Cenozoic basin. The Upper Cretaceous sequence is made of continental sandstones in the base (average thickness of 350 m) and limestones on top (0-600 m thick). The Cenozoic sequence is an up to 3500 m thick pile of sandstones and conglomerates deposited by alluvial



fans. These fans developed at the foot of the southern slope of the Cantabrian Mountains during the Alpine deformation and only the uppermost 100 m outcrop. The youngest part of the succession is horizontal, postdates the Alpine deformation and buries in many areas the trace of the frontal Alpine thrust.

It is difficult to establish the precise age of the Alpine deformation owing to the lack of good chronostratigraphical data in the mainly detritic and azoic Cenozoic sequence. However, the age of the deformation can be estimated between the Oligocene to Upper Miocene from Ostracoda fauna found near the base of the Cenozoic basin (López Olmedo et al., 1997) and mammal remains from the top of the Cenozoic succession further to the East (Portero et al., 1982). This time span coincides with the ages of uplift of the Cantabrian Mountains recorded by apatite fission-tracks to the west (Martín-González et al., 2011).

## 3. ESCIN-2 profile: description, interpretation and seismic modelling

The 65 km long deep seismic reflection profile ESCIN-2 was designed to investigate the N-S crustal structure of the southern tectonic front of the Cantabrian Mountains and the transition to the Cenozoic Duero foreland basin (Fig. 3). The northern segment of the line crossed rugged topography with altitudes between 770 an 1850 m above sea level, whilst the topography in the south is almost flat around 1000 m above sea level (datum plane 1000 m). The line was acquired with a 240-channel geophone spread, using 60 m geophone group spacing and laid out as a symmetrical split-spread with roll-on. A total of 212 single-hole dynamite shots (10-25 kg) were recorded. The record length was 25 s with a mean 30-fold coverage. More detailed information on data acquisition and processing parameters of the stack section is in Pulgar et al. (1995, 1996). The section was processed following a standard sequence and post-stack lateral coherency filtering was applied later (Fig. 4). The geological interpretation of the shallow reflections in the Duero basin is also constrained by a set of 38 oil exploration seismic reflection profiles (location in Fig. 1) that provide a clearer seismic image of the shallow levels, particularly in the transition to the Cantabrian Mountains (Gallastegui, 2000). Three oil exploration wells provided the lithological calibration of the seismic data allowing the identification of reflectors in the seismic sections (well locations in Fig. 1). The Campillo well is located only 6,5 km to the W of ESCIN-2 and 1895 m of Cenozoic and 660 m of Mesozoic were drilled before reaching the Paleozoic basement (Ordovician or Silurian).

### 3.1 *Data description and geological interpretation*

The quality of the crustal seismic image is good along the profile, although reflectivity is weaker in the central part (CDP 1000-1300), especially in the upper 4 s (CDP 900-1200 in Fig. 4). This can be attributed to bad shot quality in this area, where important lithological lateral variations and residual statics problems are evidenced by the delay in the first arrivals in the shot gathers (see shot 560, south of trace 160 in Fig. 5b).



The southern half of the profile is characterized by horizontal reflectors along the whole crust (up to 12 s) beneath the Duero basin. Reflectors are very continuous (up to tens of km) and well defined in the upper two seconds. They are especially energetic in a lower horizontal band at 1.5-2 s (A in Figs. 4, 5a and 6a) that corresponds to the upper Cretaceous rocks in the base of the sedimentary Duero basin. Reflections from the overlying Cenozoic (B) are also horizontal, but less continuous

and energetic. Both levels (A and B) are curved and slightly shifted upwards, about 0.5 s, to the North of CDP 1650. The upper crust in this domain, up to a depth of 5.5 s, is almost transparent aside from two thin bands of discontinuous and aligned north dipping reflectors that can be traced from the base of the basin to the base of the upper crust at 5.5 s where they sole (C and D in Figs. 4 and 6a). The former feature (C) cuts the upper Cretaceous and base of the Cenozoic reflectors, in coincidence with the area where they are vertically shifted, and the latter (D) fades below the sedimentary basin. Reflectivity

increases abruptly below 5.5 s in the middle and lower crust up to a depth of 12 s. The reflection Moho is laterally continuous and consistent below the Duero basin and the transition to the Cantabrian Mountains. Moho reflections are prominent and make up a 1 s thick band of subhorizontal and slightly anastomosed reflectors at 10-12 s (M in Figs. 4, 5a-b and 6d) above a less reflective upper mantle.

The seismic pattern across the crust is quite different in the northern part of the profile below the Cantabrian Mountains,

since all the crust is reflective and there is a northwards gradual increase in Moho depth and thus crustal thickness. The upper crust is reflective and dominated by near horizontal reflections, which are particularly energetic at 2 s in the northern end of the profile (E). Two parallel bands of N dipping discontinuous reflectors, similar to those described below the Duero basin, cut the upper crust and almost reach the surface (F & G) (Figs. 4 and 6b). The same N-dipping events are even more conspicuous in some of the shot gathers (for example see shot 327 in Fig. 5c). Moreover, exploration profiles, such as N03 in

Fig. 7, provide a clearer image of the nature of these structures near the surface. The north-dipping bands, interpreted in ESCIN-2 and N03, are correlated in the surface with major Variscan and Mesozoic fractures that reworked during the Alpine inversion. The horizontal middle and lower crustal reflectors observed below the Duero basin extend northwards, but gradually bend to the North and are clearly dipping northwards underneath the Cantabrian Mountains. Horizontal Moho reflections at 12 s below the Duero basin also deepen to the N and reach a maximum depth of 15 s in the northernmost end

of the profile (M in Figs. 4, 5c and 6d). In this area a wedge-shaped area of subhorizontal reflectors between 6 and 9 s (H) is truncated against the top of the N dipping package of reflectors (Figs. 4 and 6c). The mantle shows no prominent features and only short and discontinuous reflections can be traced parallel to the Moho topography (Fig. 4).

### . 3.2 *Seismic modelling results*

2D forward seismic modelling is a technique that allows to obtain a geological model by comparing its seismic response

with the real seismic data. In this case, forward seismic modelling of deep seismic reflection profile ESCIN-2 (Fig. 8) was used to i) check the theoretical ray coverage of the seismic profile, ii) support the proposed geological interpretation and iii) to obtain the depth of the different interfaces interpreted (see Gallastegui et al., 1997 for further explanation of the process).



The first modelling step was to construct a geological model of the crust, following the direction of the profile. The main reflectors and crustal levels interpreted in the seismic section and all geological and geophysical data available were included. One of the key points of the modelling technique is to precisely establish the detailed P-wave velocity-depth distribution of the model (Fig. 9b). Velocities in the model were determined from two main sources: i) the velocities of the

materials that fill the Duero basin were deduced from the three exploration wells available in the area. An interval velocity of 3.7 and 4.8 km/s was calculated for the Cenozoic and Cretaceous sequences from their respective thicknesses in the wells and the equivalent two-way travel time in exploration profiles. ii) The velocities of the materials that outcrop in the Cantabrian Mountains are consistent with measurements of similar Variscan rocks in nearby locations (Gutiérrez-Claverol et al., 1994). iii) The rest of the velocities were directly taken from a 200 km long N-S refraction profile (described in the next

section), which is coincident in the central part with ESCIN-2 (Fig. 9a) (Pulgar et al., 1996). Despite the limited resolution of refraction profiles, the refraction velocity values were considered reliable and are the only data available in this area.

In the next modelling step a synthetic seismogram (Fig. 8b) was generated by 2-D normal incidence raytracing in the velocity-depth model (Fig. 8a). It depicts the theoretical seismic response of the model and it is composed of 184 synthetic traces in a relation of 1 synthetic trace for every 11 real traces in ESCIN-2. Finally, the synthetic seismogram was compared

with the real seismic data and the initial model was gradually changed until a satisfactory fit was achieved between the real and synthetic seismic data (Fig. 8c). The raypath plot in the model (Fig. 8a) showed that there is good theoretical normal-incidence ray coverage of the different levels interpreted in the profile, giving thus a good confidence level to the interpretation. Only the lowest crustal levels in the northern end of the profile are not sampled due to their N-dipping attitude.

The crustal thickness of the final model is close to 33 km in the S under the Duero basin, where the crust can be divided in three subhorizontal levels: the upper crust, up to a depth of almost 14 km, and the middle and lower crust (Fig. 9b). The depth of the boundary between the middle and lower crust was directly taken from the refraction profile described in next section and it was included in the model in order to check the compatibility between refraction and reflection data. Crustal thickness increases to more than 47 km in the northern end of the profile under the Cantabrian Mountains. The boundary

between the upper and middle crust also deepens northwards and reaches a depth of 26 km. One of the most interesting results is that the inclined bands of reflections in the upper crust below the Duero basin and the Cantabrian Mountains dip to the N (30.5º-36º) and reach a depth of 14 km in the boundary between the upper and middle crust.

## 4. Other geophysical data

### 4.1 Seismic refraction profile

A 200 km long N-S reversed refraction profile was recorded inland to complement the seismic reflection data. This profile extends from the North Iberian coastline in the N to the center of the Duero basin in the S. It was designed to traverse the Cantabrian Mountains and is parallel to ESCIN-2 (Fig. 1). The profile provided the first seismic crustal image across the





Cantabrian Mountains (Fig. 9a) (Gallart et al., 1995; Pulgar et al., 1996; Fernández-Viejo, 1997, 2000). The most outstanding feature of the profile is the S to N crustal structural variation. The velocity-depth distribution and structure under the Duero basin are characteristic of the Variscan crust found in previous profiles along the North Iberian Variscan Massif (Córdoba et al., 1988). However, there is a northwards gradual increase in crustal thickness and an Alpine crustal root is well

developed under the Cantabrian Mountains. Moho depth is well constrained from 32 km in the South to 45 km at about 30 km inland beneath the Cantabrian Mountains. This crustal thickening has also been imaged in other N-S wide-angle experiment located about 50 km to the W that extends northwards in the Cantabrian Margin (Pulgar et al., 1996; Fernández-Viejo, 1997; Fernández-Viejo et al., 1998). Nearby E-W refraction profiles show that the crustal root trends E-W and disappears westwards in the continental margin (Fernández-Viejo, 1997; Fernández-Viejo et al., 2000) and extends

eastwards under the Basque-Cantabrian basin in continuity with the Pyrenean crustal root (Pedreira et al., 2003, 2007).

### 4.2 Gravity profile

To further constrain the interpretation of the seismic data a 2-D gravity profile was modeled (Fig. 9d-e). A Bouguer anomaly map was compiled with data supplied by the BGI (Bureau Gravimétrique International) and complemented with new measurements collected for this research (Fig. 10). Data are referenced to the IGSN-71, converted to Free Air anomalies

using the Geodetic Reference System Formula (GRS-67) and to simple Bouguer anomalies with a density of 2670 $kg/m^3$. Terrain corrections were computed to a distance of 20 km around each measured point. The Bouguer anomaly shows an E-W elongated gravity low located to the W of the southern end of ESCIN-2 that reaches values around -90 mGal and extends to the end of ESCIN-2, where the anomaly has a value of -75 mGal. The anomaly low superposes the area with the thickest sedimentary fill in the northern part of the Duero basin. Bouguer anomaly values gradually increase northwards towards the

coastline, where anomaly values range from 0 to 30 mGal.

Depths of the interfaces to construct the model were directly taken from seismic data and densities of the crustal levels were calculated from P-wave velocities obtained from the refraction profile parallel to ESCIN-2. The chosen densities were: 2670-2700 $kg/m^3$ for the upper crust, 2750 $kg/m^3$ for the middle crust, 2850-2900 $kg/m^3$ for the lower crust and 3300 $kg/m^3$ for the upper part of the mantle. The densities of the Mesozoic rocks in the Duero basin (2620 $kg/m^3$) were converted from the

velocities defined previously from well data and the densities of the Cenozoic levels decrease southwards from 2600 to 2250 $kg/m^3$ because of the change to distal and finer-grained facies in that direction.

2D gravity modelling of the data evidences that the density model and the structure deduced from geological and seismic data can account for the observed gravity anomalies. Moho depths are in agreement with those derived from other geophysical methods, increasing from 32 km under the Duero basin to almost 50 km under the Cantabrian Mountains. The

gravity low in the south is related to the sedimentary fill of the Duero basin and anomaly values increase northwards due to the combined effect of i) the superposition of the northern lower and middle crust above the crustal root and ii) the thinning of the crust towards the North Iberian continental margin. This thinning is not shown in Fig. 9d since it occurs further to the



N, but its gravity effect is included in the profile (Fig. 9e). Both effects contribute to mask the expected anomaly low that the crustal root would produce.

### 4.3 Magnetotelluric profile

Pous et al. (2001) built a 2D conductivity model, almost coincident with ESCIN-2, from the coastline to the Duero basin. There is a remarkable agreement between the interpreted structures in the seismic section and in the magnetotelluric model (Fig. 9f). In the upper crust several north dipping high electrical conductivity zones reach a depth of 15 km. The southernmost one is coincident with dipping reflections F and G interpreted in ESCIN-2. Further north, between the northern end of the seismic profile and the coast line, other north dipping conductors image at depth the prolongation of two major outcropping Alpine faults (Cabuerniga and San Vicente de la Barquera faults) parallel to the structures shown in the reflection profile. There are also coincidences at deeper crustal levels. First, there is wedge-shaped conductive zone in the same position as the wedge of reflectors named H and second, the MT model has also shown the crustal thickening under the Cantabrian Mountains, which was revealed by the rest of the geophysical data.

### 5. Geological interpretation and discussion

One of the distinctive seismic features of the studied area is the change in the reflectivity pattern and the crustal structure between the southern and northern part (Figs. 4b and 11). The southern part (below the Duero basin) is characterized by poor reflectivity in the almost transparent upper crust (down to 5-6 seconds or 14 km depth) and an abrupt increase in reflectivity below. The high reflectivity is persistent in the reflective lower crust up to a depth of 10-12 s (approximately 32 km) where there is a sharp contrast between a conspicuous reflective Moho and a low reflective upper mantle. On the other hand, the whole crust is reflective in the northern domain (below the Cantabrian Mountains) and the crust is thickened to 15 s (about 45 km) as a result of an important Alpine reworking of the crust at all crustal levels.

The interpretations of the commercial seismic reflection profiles and ESCIN-2 show that the Duero basin was only slightly deformed in the Alpine deformation event (Fig. 11). The only structures that evidence the compressional deformation are found: i) in the northern border where the Mesozoic and base of the Cenozoic are overturned and a syntectonic unconformity developed in the upper part of the Cenozoic succession (Alonso et al., 1996) and ii) 14 km to the south of the mountain front where a buried progressive unconformity and a minor uplift (Campillo uplift) and drape fold developed (Figs. 4 and 6a). The two parallel band of reflectors located below the Duero basin (bands C and D in ESCIN-2) are interpreted as basement thrust that seem to sole in the transition between the upper and middle crust at 14 km. Band C corresponds to the buried Campillo thrust responsible for the deformation of the Cenozoic rocks and the Campillo uplift in its hanging-wall, that are imaged in ESCIN-2 and other commercial seismic reflection profiles. Band D is a secondary parallel thrust, which fades out upwards and did not displace the base of the basin.



The Alpine deformation was more intense below the Cantabrian Mountains. The N-dipping bands of reflectors that cut across the upper crust in ESCIN-2 are the seismic expression of the compressional structures responsible for this deformation. These basement faults have also been interpreted by the magnetotelluric method (Pous et al., 2001). Band G is the seismic image of the main Alpine thrust, previously interpreted by Alonso et al. (1996) from geological data. According

to these authors, the southwards displacement along this basement involved-thrust produced the uplift of the Cantabrian Mountains and the overturning of the northern border of the Duero basin. Parallel band F matches in surface with the Mesozoic extensional Ruesga-Ubierna Fault (Fig. 3) that also reworked as a thrust in Alpine times. Both thrusts cut across the upper crust up to a depth of 14 km where they seem to sole. The main Alpine thrust shows no associated prominent reflections in the upper 2 s in ESCIN-2, however, this upper part has been clearly imaged in the commercial reflection

profiles, together with other N-dipping thrusts located further to the N that coincide in the surface with Variscan structures that reworked during the Alpine event. (Profile N03 in Fig. 7). The dip of the crustal thrust deduced from reflection data is 30°-36°, which is significantly different than the one interpreted for the main Alpine thrust from geological data (16°). The same steep angle was found in a similar crustal scale thrust interpreted in COCORP Wind River profile (Wyoming) (Smithson et al., 1979).

One of the most striking features in the profile is the crustal thickening beneath the Cantabrian Mountains revealed by the deepening of the Moho from 33 km (12 s) in the S to 47 km (15 s) in the N. The maximum Moho depth has been established in 55 km below the coastline by refraction/wide angle modelling. The deepening is progressive as a consequence of the simultaneous bending of the middle and lower crust. The top of the middle crust gets deeper from 14 km below the Duero basin to about 26 km in the N. The wedge of reflectors (H) plays a major role in the interpretation of the Alpine structure of

the profile and has to be considered in a broader regional perspective. The lower crust of the Cantabrian margin is detached and was underthrusted to the S under the North Iberian margin due to the alpine deformation that affected the area. This lower crust deeply indented to the S into the Iberian crust, below the Cantabrian Mountains, and split the latter, whose lower portion (middle and lower crust) was forced to subduct to the N building a crustal root. The wedge of reflectors (H) represents the southernmost end of the Margin lower crust, which also forced thrust emplacement and the Alpine uplift of the

Cantabrian Mountains. The overall crustal structure is very similar to that of other Alpine orogens such as the Alps or the Pyrenean range. The later constitutes the eastern continuation of the Cantabrian Mountains and was built in the same tectonic event. In the Pyrenees the Iberian plate is also indented by a wedge of its northern counterpart plate and is forced to subduct northwards creating a crustal root. One key difference between both mountain ranges is that the later is a collisional orogen, whilst there was no collision in the Cantabrian Mountains. However, the continuity between both Alpine roots indicates that

the overall crustal structure of the Cantabrian Mountains may have been conditioned and driven by the initial development of the Pyrenean root at the end of Cretaceous times and the westward migration of the deformation towards the studied area.



## 6. Conclusions

Deep seismic reflection profiling in the Southern branch of the Cantabrian Mountains and the transition to the Cenozoic Duero basin has provided a good image of the crustal structure (Fig. 11). The crust exhibits N-S lateral variations that result from the superposition of the Alpine orogenic deformation (Cenozoic) on a Variscan structured crust. The crust below the foreland Duero basin is almost undeformed and shows a reflective pattern that is common for most of the undeformed Caledonian and Variscan crusts in Europe, according to Mooney and Meissner (1992). An almost transparent upper crust, below the slightly deformed Mesozoic and Cenozoic deposits that fill the Duero basin, overlies a reflective lower crust from 14 to 32 km, where an energetic reflective Moho leads to a poor reflective upper mantle. However, as a result of the Alpine orogeny, that led to the building of the Pyrenees further to the East, the crustal reflectivity pattern changes significantly below the Cantabrian Mountains. The whole crust is reflective as a result of the Alpine reworking. The main shallow structures are a set of north dipping basement thrusts (30°-36°) that cut the upper crust and sole at 14 km in a common detachment. The most important one is the Main Alpine Thrust, whose propagation fold gives shape to the northern boundary of the Duero basin and is responsible for the Alpine uplift of the Cantabrian mountains. Other crustal thrusts interpreted in the North in ESCIN-2, and several commercial reflection profiles, coincide in surface with major Variscan and/or Mesozoic structures that reworked in Alpine times.

Another result of the Alpine deformation, shown in seismic profiles (ESCIN-2 and refraction experiments) and confirmed by other geophysical methods (gravity and MT modelling) is that the crustal thickness gradually increases northwards below the Cantabrian Mountains, reaching a maximum of 55 km beneath the coastline. We have interpreted that there is a process of tectonic wedging and duplication of the crust as a result of the southwards indentation of the lower crust of the Iberian Margin, which forced the delamination and northwards subduction of the Iberian crust at deep crustal levels. This process drove the emplacement of the aforementioned thrusts in the upper crust. The overall crustal structure is similar to the one found in the Pyrenees where an Alpine crustal root developed as a result of the collision between two continental plates. In fact, although both mountain ranges are separated by the Basque-Cantabrian basin, the Alpine crustal root is continuous between them, making them a unique and continuous orogen at the crustal scale. The development of the Pyrenean root in the upper Cretaceous and the progressive westwards migration of the Alpine deformation from the Pyrenean area to the Cantabrian Mountains conditioned the structure in the latter, where there was N-S plate convergence, but not collision.

*Acknowledgements*. This study was part of the PhD thesis of J. Gallastegui and was supported by a FPU grant and research projects GEO 90-0660-1086 and PB92-1013 funded by CICYT (Committee of Science and Technology of the Spanish Ministry of Education and Science; project) and FICYT (Foundation for the Science and Technologic Research, Government of Asturias, Spain). Part of the study has also been financed by the Spanish Ministry of Science through the "TOPOIBERIA" Consolider Project (ref: MEC-06-CSD2006-0041) and the MISTERIOS Project (ref: MINECO-13-CGL2013-48601-C2). We gratefully acknowledge the Bureau Gravimétrique International (BGI) for providing part of the gravity data and REPSOL for providing commercial seismic reflection profiles.





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

**Figures**

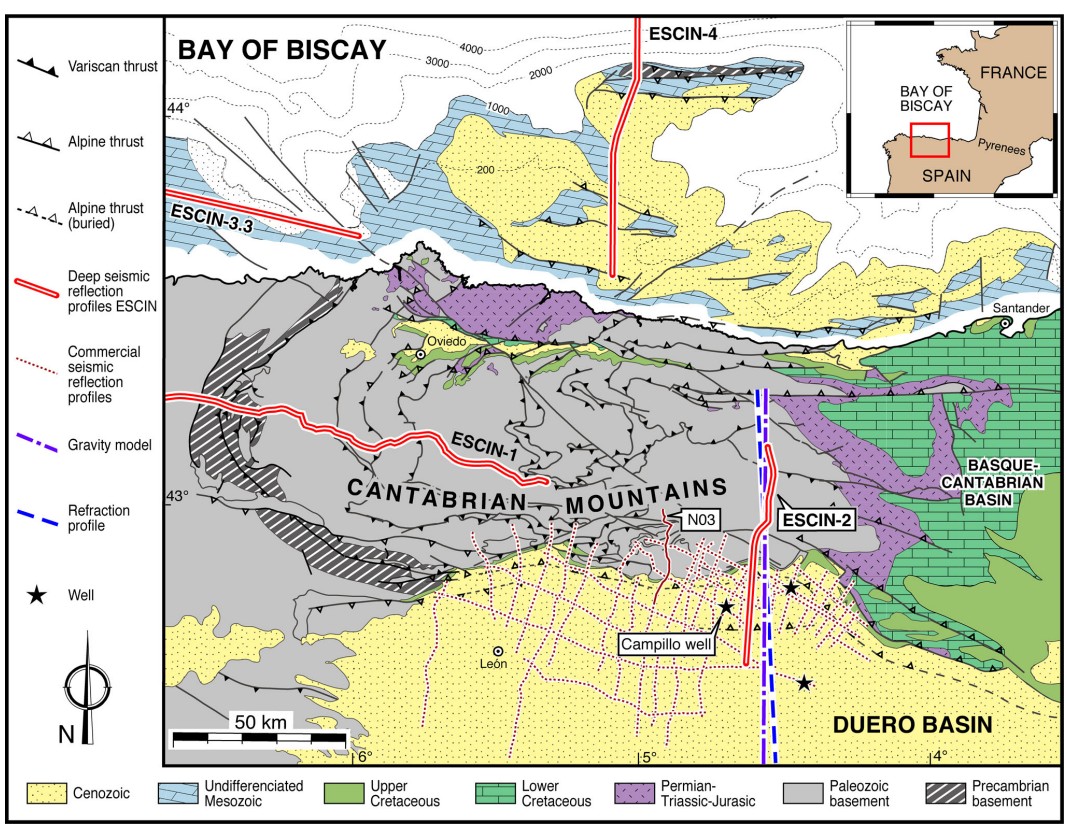

Figure 1. Simplified geological map of NW Iberia and its continental platform. Location of deep seismic reflection profile ESCIN-2 and the rest of the geophysical data mentioned in the text.





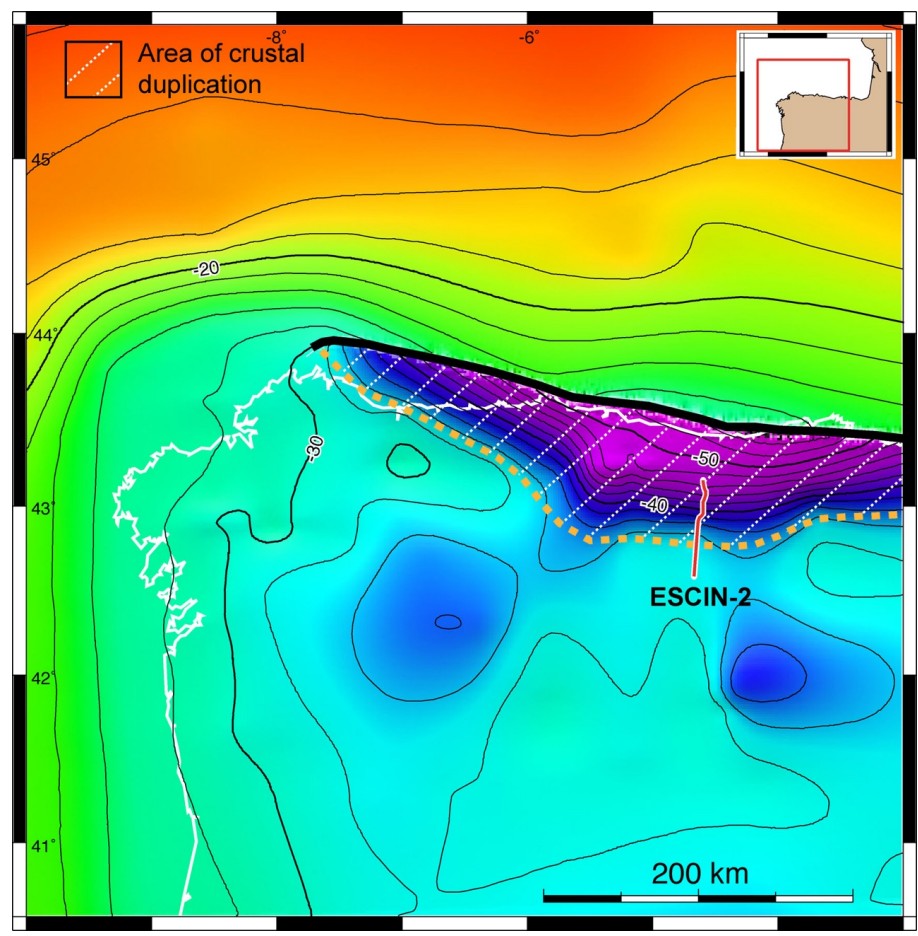

Figure 2. Interpolated crustal depth model for the NW of the Iberian Peninsula from Gallastegui (2000). The dashed area shows the extent of the area of crustal indentation. Moho depth contours are represented every 2 km.

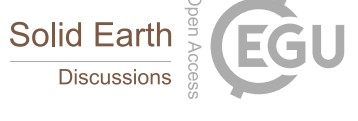

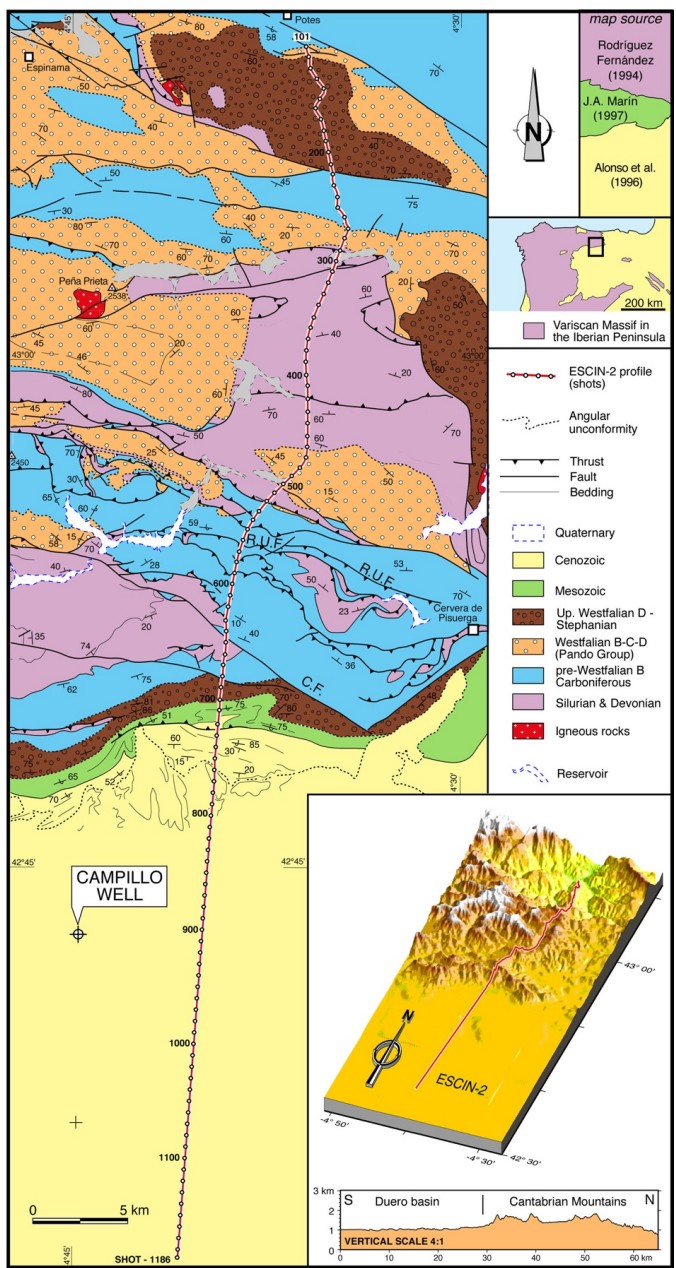



Figure 3. Geological map of the studied area. The inset box in the lower right-hand corner shows 3-D DTM view of the region and the elevations profile along ESCIN-2 that shows the flat topography in the Duero basin in contrast with the rugged mountainous terrain in the Cantabrian Mountains.

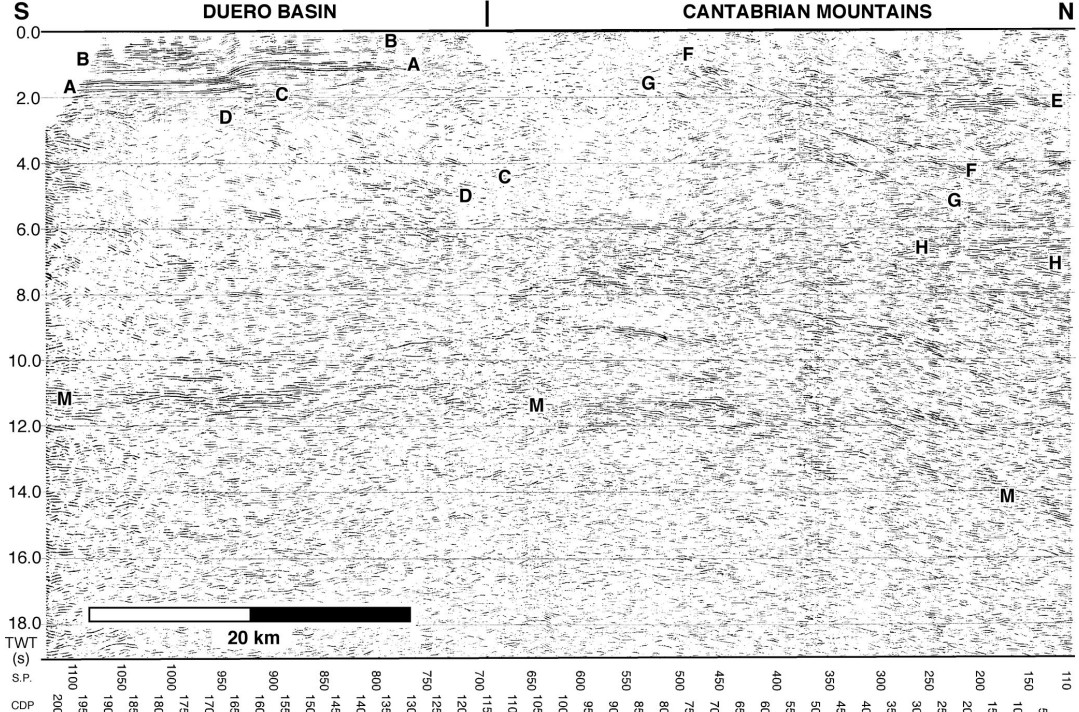

5    Figure 4. Coherency filtered stack section of ESCIN-2 with the interpretation of the main reflectors and crustal levels described in the text.





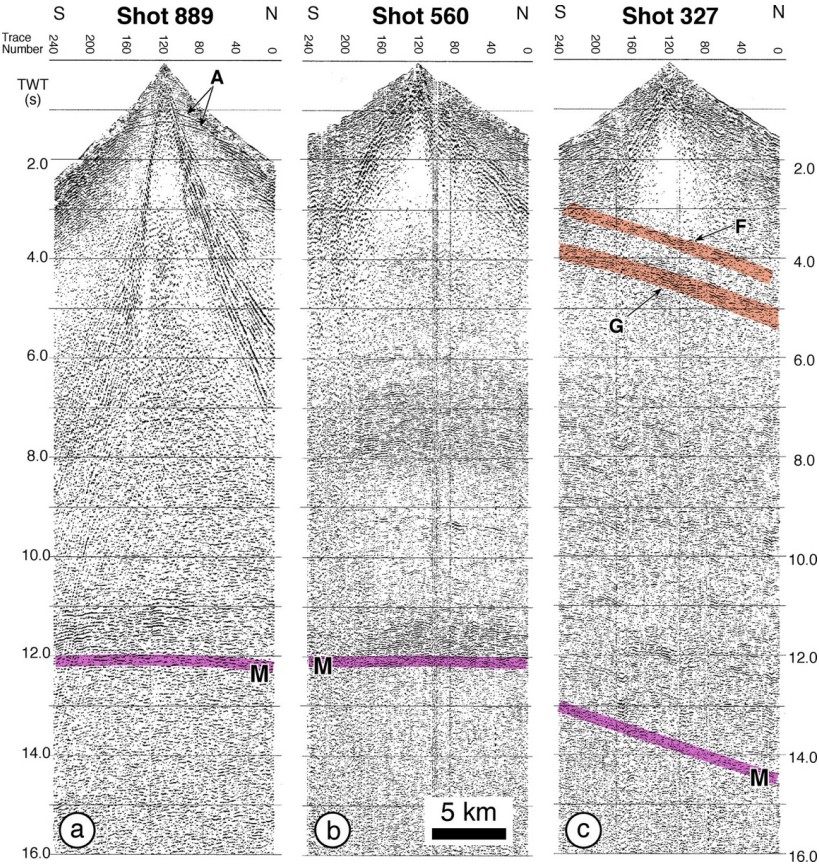

Figure 5. Three shot-gathers from ESCIN-2 deep seismic reflection profile. a) Shot-gather from the Duero basin (A-reflections from the Cretaceous levels that drape the bottom of the basin). b) Shot gather from the transition to the Cantabrian Mountains, note the increase in reflectivity below 6 s and the base of horizontal Moho (M) at 12 s. c) Shot gather from the Cantabrian Mountains. North dipping reflecting bands are more conspicuous than in the stack section (F and G). Moho is dipping to the N and reaches a maximum depth of almost 15 s.



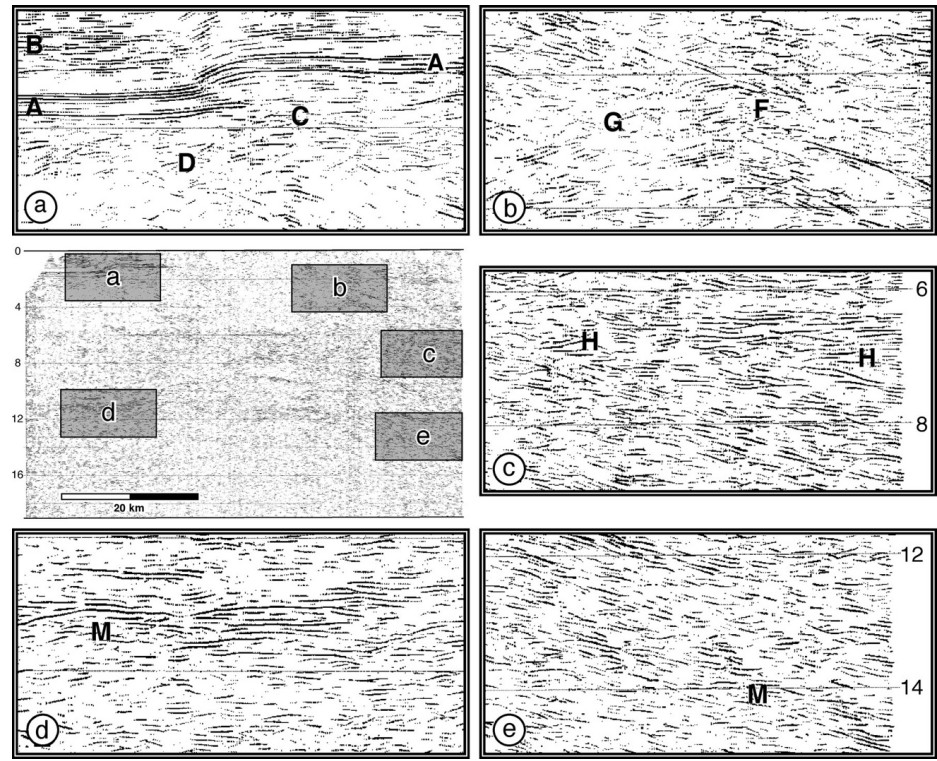

Figure 6. Magnification windows from selected regions of ESCIN-2. a) Campillo thrust (C) and uplift in the Duero basin. b) North dipping band of reflectors interpreted as crustal scale thrusts (F & G). c) Wedge of subhorizontal reflectors indented in the crust (H). d) Horizontal reflection Moho (M) at the base of the crust (11 s) beneath the Duero basin. e) North dipping reflection Moho (M) at the base of the crust reaching a depth of 15 s below the Cantabrian Mountains.



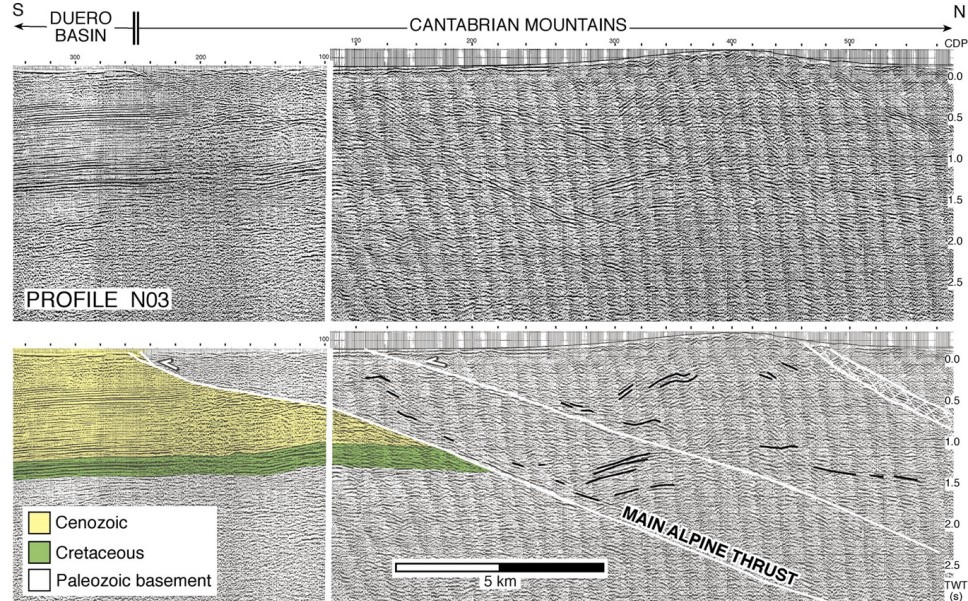

Figure 7. Commercial seismic reflection profile N03 across the northern border of the Duero basin (location in Fig. 1). Note the N-dipping reflectors that can be correlated with reflective bands F & G in ESCIN-2. The southernmost one corresponds to the main alpine thrust in the border of the basin and the northern one coincides in the surface with a major Variscan structure reworked in Alpine times.



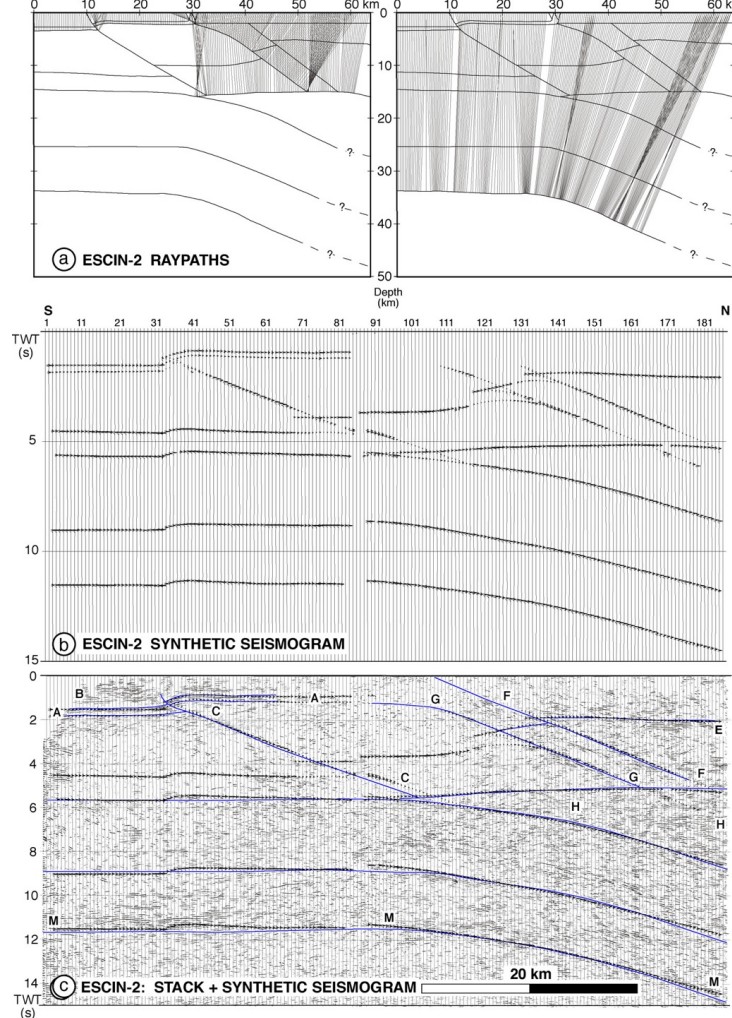

Figure 8. a) Ray-path plot of normal incidence raytracing to different interfaces in the final ESCIN-2 velocity depth model in Fig. 9b. Note the areas with no theoretical ray coverage under the Cantabrian Mountains. b) Synthetic seismogram obtained from normal incidence raytracing showing the seismic response of the same velocity-depth model. c) Synthetic seismogram of the final model superimposed on ESCIN-2 and the boundaries of the velocity-depth model. It shows a good match between the real and synthetic seismic data.





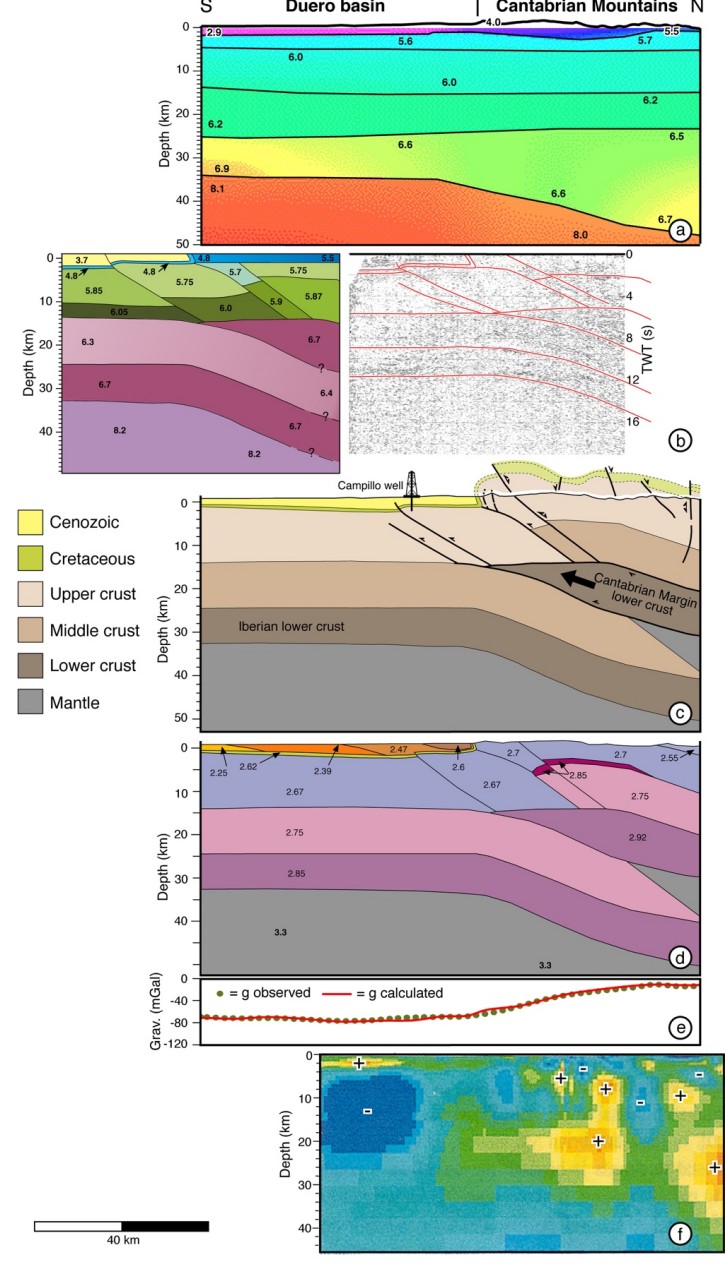



Figure 9. a) Refraction velocity model parallel to ESCIN-2. (Pulgar et al., 1996; Fernández-Viejo, 1997; Gallart et al., 1997). b) Velocity-depth model of ESCIN-2 obtained by forward seismic modelling of interphases shown in red in the interpreted deep seismic section. c) Geological cross-section of the area that synthesizes all information available. d & e) 2-D Gravity model, observed and calculated gravity anomaly of the studied area (Gallastegui, 2000). f) 2-D electrical resistivity model and location of resistivity highs and lows (adapted from Pous et al., 2001).

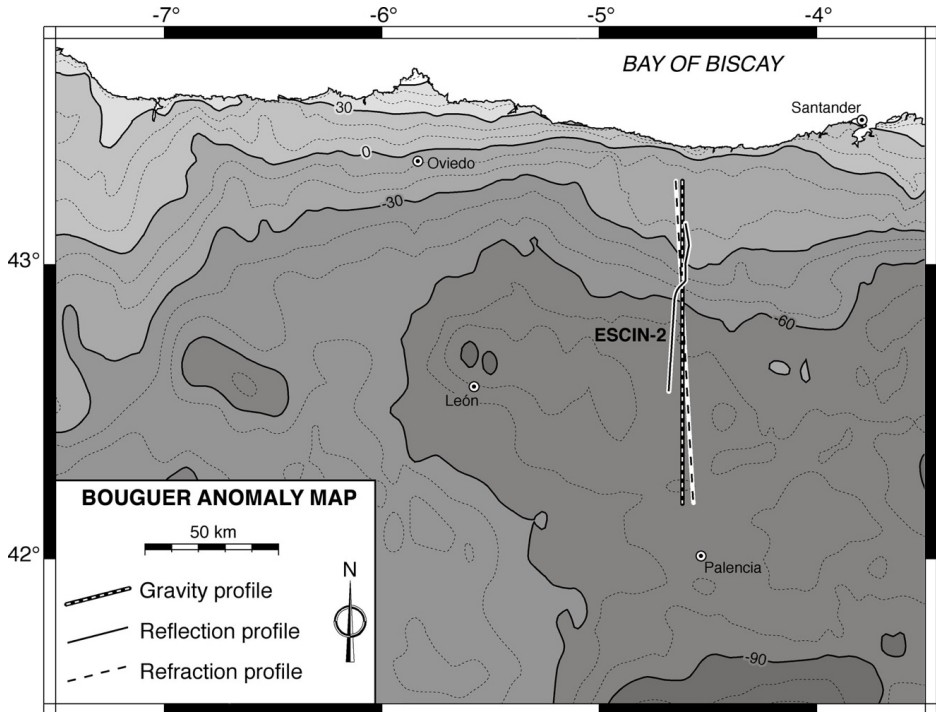

Figure 10. Bouguer anomaly map compiled with data supplied by the BGI (Bureau Gravimétrique International) and complemented with new measurements collected for this study.









Figure 11. a) Interpreted stack section of ESCIN-2. b) Geological interpretation of ESCIN-2. The base of the crust deepens from almost 33 km below the Duero basin to 47 km under the Cantabrian Mountains due to the indentation of the Cantabrian Margin lower crust into the detached Iberian crust.