# Peer review of "Alpine tectonic wedging and crustal delamination in the Cantabrian Mountains (NW Spain)"

_Solid Earth, 2016_

## Referee Comment (RC1) · Anonymous Referee #1 · 14 Apr 2016

Review of manuscript: Alpine tectonic wedging and crustal delamination in the Cantabrian Mountains (NW Spain) Jorge Gallastegui, Javier A. Pulgar, Josep Gallart

The manuscript focuses on the deep crustal structure of the southern part of the Cantabrian Mountains and the transition to the adjacent Cenozoic Duero basin imaged by the deep seismic reflection profile ESCIN-2. The authors present a geological cross-section of the area integrating published data (ESCIN-2, a refraction profile and a conductivity model) with a depth model of ESCIN-2 obtained by 2-D seismic modelling, a gravity model coincident with ESCIN-2 and all geological/geophysical data available (wide-angle/refraction and MT data). The manuscript provides a crustal scale view across one part of the Cantabrian Mountains, which is valuable, however, the approach is very technical. Indeed, the study would benefit from a more developed discussion linking observations/modelling done on one section with the geological evolution of the

whole Cantabrian/Pyrenean mountain belt.

Detailed comments: l.10 p.1 dip of the thrust is 30 to 36°; what is the geometry of the thrust? Is it a ramp or does it form a ramp flat structure? It would be interesting to have a more detailed description of the fault system.

l.10 p.2 reactivation of older Variscan structures: what is the criteria to demonstrate that the structures are Variscan?

l.20 p.6 lower and upper crust: how can they be recognized, based on what criteria?

l.25 p.6 boundary between upper and middle crust: how defined? What is the nature of this interface?

l.25 p.8 What is "Campillo uplift"?

l.30 p.8 Band D (how interpreted)

l.30 p.9 The comparison/differences with the Pyrenees (and other chains?) need to be better developed and can not be based only on statements.

l.20 p.10 Crustal roots of the Pyrenees are connected to those of the Cantabrian Mountains (this is neither shown nor is there a reference that supports this statement).

l.25 p.10 Westward migration of the Alpine deformation: what is the evidence? Provide either observations or references

---

## Referee Comment (RC2) · Anonymous Referee #2 · 26 Apr 2016

This paper presents an unified interpretation of seismic reflection, refraction, gravity and MT data beneath the Cantabrian Mountains in Spain and adjacent basin. The paper is relevant and appropriate for inclusion in Solid Earth with minor revision. General comments: (1) The data description beginning on page 4 line 27 is hard to follow. Consider moving much of the detailed references to CDP numbers and reflector times to the figure captions (2)This paper is generally well written; however, consider replacing the many indefinite "it"s with the noun that they refer to. Also, you should consider rewriting the any sentence that starts with "There is" and bring the noun to the front. These suggestions will make your meaning more clear. (3) Page 6 line 11: Why are the velocities from the refraction profiles consider reliable? Attached is a scan of my hand‐written editorial comments.

Please also note the supplement to this comment:
http://www.solid-earth-discuss.net/se-2016-23/se-2016-23-RC2-supplement.pdf

[Figure]

**Supplement:**

Review:

*Alpine tectonic wedging and crustal delamination in the Cantabrian Mountains (NW Spain)*

by Callastegui et al.

This paper presents an unified interpretation of seismic reflection, refraction, gravity and MT data beneath the Cantabrian Mountains in Spain and adjacent basin. The paper is relevant and appropriate for inclusion in Solid Earth with minor revision.

General comments:

(1) The data description beginning on page 4 line 27 is hard to follow. Consider moving much of the detailed references to CDP numbers and reflector times to the figure captions

 (2)This paper is generally well written; however, consider replacing the many indefinite "it"s with the noun that they refer to. Also, you should consider rewriting the any sentence that starts with "There is" and bring the noun to the front. These suggestions will make your meaning more clear.

(3) Page 6 line 11: Why are the velocities from the refraction profiles consider reliable?

Attached is a scan of my hand-written editorial comments.

[revised manuscript text omitted]

---

## Author Comment (AC1) · 24 May 2016

We would like to express our gratitude to the anonymous reviewer (#1). for his thorough review. The constructive comments helped to improve the manuscript. In this file, we will reply the detailed comments included in the revision. All changes will be added to the last version of the manuscript. We attach a commented pdf version of the final text after the revision from both reviewers as a supplement file. Lines and pages in our reply refer to those in the supplement file.

REPLY TO COMMENTS REFEREE 1

l.10 p.1 dip of the thrust is 30 to 36; what is the geometry of the thrust? Is it a ramp or does it form a ramp flat structure? It would be interesting to have a more detailed description of the fault system.

[Figure]

We agree with your comment and have added in l.14 p1 that thrusts have a "ramp geometry and sole in the boundary with the middle crust."

l.10 p.2 reactivation of older Variscan structures: what is the criteria to demonstrate that the structures are Variscan?

This statement is taken directly from the two references in the text (Alonso et al., 1996, Pulgar et al., 1999) and is a regional scale conclusion derived from geological maps and outcrops. These authors base their conclusion on observations made in nearby areas with Mesozoic and Cenozoic outcrops such as: 1) The boundary between the Variscan Cantabrian Zone and the Mesozoic Basque-Cantabrian basin where the alpine reactivations of older Variscan structures are affecting the Mesozoic units. This reactivation is evident in the map in figure 1 to the east of ESCIN-2 profile. 2) The deformation of the Meso-Cenozoic Oviedo Basin, to the west of ESCIN-2 profile, which is located in figure 1 but not mentioned in our text and also the deformation along the coastal section. In areas where there are no Mesozoic levels involved in the alpine deformation it is difficult or even impossible to evaluate whether there is any alpine deformation superposed to the Variscan structures or its amount.

l.20 p.6 lower and upper crust: how can they be recognized, based on what criteria?

They can be recognized in terms of differences in reflectivity. We explain the upper-middle crust boundary in section 3.1 (l.8-13 p.5). The upper crust is poorly reflective and extends to 5.5 s, where there is an increase of reflectivity interpreted as the top of the middle crust. The middle-lower crust boundary is not evident in the reflection profile and that is why we don't refer to it in the description. It was included in our model at the depth interpreted in the refraction profile. It is explained in l.24 p.6 that this boundary "was directly taken from the refraction profile described in next section and it was included in the model in order to check the compatibility between refraction and reflection data." We have added the word "reflective" in l.12 p.5 to emphasize that the whole middle and lower crust are reflective. We have interpreted the lowermost reflective band at 10-12 s as the reflection Moho in the base of the middle-lower reflective crust.

l.25 p.6 boundary between upper and middle crust: how defined? What is the nature of this interface?

We have interpreted this boundary at 5.5 s in the stack section (l.8 p.5) which is equivalent to 14 km in the modelled section (l.13 p.9). This boundary is interpreted at that depth where there is an important increase in reflectivity and it is coincident in depth with the boundary between the upper and middle crust in the refraction profile (Fig. 9a). It is also the boundary where the upper crust thrusts sole (l.13 p.9) and the local depth of the seismic zone (we have added a reference from Llana-Fúnez and López-Fernández, 2015). We don't know the reason for the reflectivity increase or the nature of this boundary. Articles that deal with the refraction experiment do not argue the nature of the boundary either.

l.25 p.8 What is "Campillo uplift"?

The Campillo uplift is the uplift of the Meso-Cenozoic succession in the hanging wall of the crustal thrust that we name C (Figs 4 and 6a). We have included a longer sentence in l.31 p-8 to explain it and added a reference and named it as "c.u." in figure 6a and its figure-caption l.15 p.16.

l.30 p.8 Band D (how interpreted)

Band D is interpreted from a series of N-dipping short reflections, parallel to band C, which is more conspicuous. In this sense we have added this characteristic in l.11 p.5 ". . .and the latter (D) is less conspicuous and fades. . .". We have to say that this band of reflections is more evident in larger plots of ESCIN-2, but unfortunately it is slightly less visible in the reduced figures built for publication. Anyway, we think that it can be seen in figure 6.a.

l.30 p.9 The comparison/differences with the Pyrenees (and other chains?) need to be

better developed and can not be based only on statements.

We refer to the similarities of the Pyrenees and Cantabrian Mountains in the introduction (section 1) and in the discussion (section 5). In the final part of the introduction we give references to articles that deal with the similarity and continuity of the crustal thickening between the Pyrenees and Cantabrian Mountains (added one more) and in the discussion we describe briefly the structure of the Pyrenees after the description of the structure in the Cantabrian Mountains. So we think that we base the comparison and differences on results from other experiments described in the references included. Any reader can refer to those articles for more detailed descriptions. We have deleted the reference to the Alps because it is out of the scope of this article to compare it with another mountain belts.

l.20 p.10 Crustal roots of the Pyrenees are connected to those of the Cantabrian Mountains (this is neither shown nor is there a reference that supports this statement).

We think that this statement is well supported by a number of references in the text: 1) In section 1 we give three references of studies that have demonstrated this continuity, based on different experiments (l.29-p.2: "Pedreira et al. (2003, 2007) and Díaz et al. (2012) proved that this structure extends eastwards..."). 2) In l33-p.2 there are two more references, in the original text, and we have rewritten the sentence to make it clearer "Crustal depth models (Fig. 2) compiled from deep sounding experiments by Gallastegui (2000) and by Díaz and Gallart (2009) also show: i) the crustal thickening, with Moho depths up to 50 km in the NW of the Iberian Peninsula and ii) the continuity of this E-W crustal structure from the Pyrenees to the Cantabrian Mountains".

l.25 p.10 Westward migration of the Alpine deformation: what is the evidence? Provide either observations or references

Thank you for the comment, we did not include references to support this statement. We have added references from three studies in section 5 (l.3-6 p.10) that have discussed the westwards migration of the onset of deformation from the Pyrenees to

the studied area and even further west, in the westernmost areas of the Cantabrian Mountains: 1) Teixell, 1998, 2) Gallastegui, 2000, 3) Martín-González et al 2014.

Please also note the supplement to this comment:
http://www.solid-earth-discuss.net/se-2016-23/se-2016-23-AC1-supplement.pdf

**Supplement:**

[revised manuscript text omitted]


this area, where important lithological lateral variations and residual statics problems are evidenced by the delay in the first arrivals in the shot gathers. (See the southern part of shot 560 in Fig. 5b.)

The southern half of the profile is characterized by horizontal reflectors at all crustal levels (up to 12 s) beneath the Duero basin. Reflectors are very continuous (up to tens of km) and well defined in the upper two seconds. Amplitudes are

- 5 especially high in a lower horizontal band at 1.5-2 s (A in Figs. 4, 5a and 6a) that corresponds to the upper Cretaceous rocks in the base of the sedimentary Duero basin. Reflections from the overlying Cenozoic (B) are also horizontal, but less continuous and with less amplitude. Both levels (A and B) are curved and slightly shifted upwards (about 0.5 s) to the North of the basin. The upper crust beneath the basin, up to a depth of 5.5 s, is almost transparent aside from two thin bands of discontinuous and aligned north dipping reflectors that can be traced from the base of the basin to the base of the upper crust
- 10 at 5.5 s where they sole (C and D in Figs. 4 and 6a). The former feature (C) cuts the upper Cretaceous and base of the Cenozoic reflectors, in coincidence with the area where they are vertically shifted, and the latter (D) is less conspicuous and fades below the sedimentary basin. Reflectivity increases abruptly below 5.5 s in the reflective middle and lower crust up to a depth of 12 s. The reflection Moho is continuous and consistent below the Duero basin and the transition to the Cantabrian Mountains. Moho reflections are prominent and make up a 1 s thick band of subhorizontal and slightly anastomosed reflectors at 10-12 s (M in Figs. 4, 5a-b and 6d) above a less reflective upper mantle.
- The seismic pattern across the crust is quite different in the northern part of the profile below the Cantabrian Mountains, since all the crust is reflective and Moho depth and crustal thickness gradually increase northwards. The upper crust is reflective and dominated by near horizontal reflections, which are particularly energetic at 2 s in the northern end of the profile (E). Two parallel bands of N dipping discontinuous reflectors, similar to those described below the Duero basin, cut
- 20 the upper crust and almost reach the surface (F & G) (Figs. 4 and 6b). The same N-dipping events are even more conspicuous in some of the shot gathers (for example see shot 327 in Fig. 5c). Moreover, exploration profiles, such as N03 in Fig. 7, provide a clearer image of the nature of these structures near the surface. The north-dipping bands, interpreted in ESCIN-2 and N03, are correlated in the surface with major Variscan and Mesozoic fractures that reworked during the Alpine inversion. The horizontal middle and lower crustal reflectors observed below the Duero basin extend northwards, but
- 25 gradually bend to the North and are clearly dipping northwards underneath the Cantabrian Mountains. Horizontal Moho reflections at 12 s below the Duero basin also deepen to the N and reach a maximum depth of 15 s in the northernmost end of the profile (M in Figs. 4, 5c and 6d). In this area a wedge-shaped area of subhorizontal reflectors between 6 and 9 s (H) is truncated against the top of the N dipping package of reflectors (Figs. 4 and 6c). The mantle shows no prominent features and only short and discontinuous reflections can be traced parallel to the Moho topography (Fig. 4).

**30 3.2 Seismic modelling results**

2D forward seismic modelling produces a geological model by comparing its seismic response with the real seismic data. In this case, forward seismic modelling of deep seismic reflection profile ESCIN-2 (Fig. 8) was used to i) check the theoretical

|---|----------------------------------|

ray coverage of the seismic profile, ii) support the proposed geological interpretation and iii) to obtain the depth of the different interfaces interpreted. (Eurther explanation of the process in Gallastegui et al., 1997.),

The first modelling step was to construct a geological model of the crust, following the direction of the profile. The main reflectors and crustal levels interpreted in the seismic section and all geological and geophysical data available were

- 5 included. One of the key points of the modelling technique is to precisely establish the detailed P-wave velocity-depth distribution of the model (Fig. 9b). Velocities in the model were determined from two main sources: i) the velocities of the materials that fill the Duero basin were deduced from the three exploration wells available in the area. An interval velocity of 3.7 and 4.8 km/s was calculated for the Cenozoic and Cretaceous sequences from their respective thicknesses in the wells and the equivalent two-way travel time in exploration profiles. ii) The velocities of the materials that outcrop in the
- 10 Cantabrian Mountains are consistent with measurements of similar Variscan rocks in nearby locations (Gutiérrez-Claverol et al., 1994). iii) The rest of the velocities were directly taken from a 200 km long N-S refraction profile (described in the next section), which is coincident in the central part with ESCIN-2 (Fig. 9a) (Pulgar et al., 1996). Despite the limited resolution of refraction profiles, the refraction velocity values were used as they are the only data available in this area.
  A synthetic seismogram of the profile (Fig. 8b) was generated by 2-D normal incidence raytracing in the velocity-depth
- 15 model (Fig. 8a) in the next modelling step. The synthetic profile depicts the theoretical seismic response of the model It is composed of 184 synthetic traces in a relation of 1 synthetic trace for every 11 real traces in ESCIN-2. Finally, the synthetic seismogram profile was compared with the real seismic data and the initial model was gradually changed until a satisfactory fit was achieved between the real and synthetic seismic data (Fig. 8c). The raypath plot in the model (Fig. 8a) showed the good theoretical normal-incidence ray coverage of the different levels interpreted in the profile, giving thus a good
- 20 confidence level to the interpretation. Only the lowest crustal levels in the northern end of the profile are not sampled due to their N-dipping attitude.

The crustal thickness of the final model is close to 33 km in the S under the Duero basin, where the crust can be divided in three subhorizontal levels: the upper crust, up to a depth of almost 14 km, and the middle and lower crust (Fig. 9b). The depth of the boundary between the middle and lower crust was directly taken from the refraction profile described in next

25 section and it was included in the model in order to check the compatibility between refraction and reflection data. Crustal thickness increases to more than 47 km in the northern end of the profile under the Cantabrian Mountains. The boundary between the upper and middle crust also deepens northwards and reaches a depth of 26 km. One of the most interesting results is that the inclined bands of reflections in the upper crust below the Duero basin and the Cantabrian Mountains dip to the N (30.5°-36°) and reach a depth of 14 km in the boundary between the upper and middle crust.

|---------------------------------------------|

|----------------------------------------|

**4. Other geophysical data**

**4.1 Seismic refraction profile**

[revised manuscript text omitted]

| Eliminado | the Alps or |
|-----------|-------------|
| Eliminado | later       |
| Eliminado | was built   |
| Eliminado | later       |

collisional orogen, whilst no collision occurred in the Cantabrian Mountains. However, the continuity between both Alpine roots indicates that the overall crustal structure of the Cantabrian Mountains may have been conditioned and driven by the onset of deformation and initial development of the Pyrenean root at the end of Cretaceous times (Teixell, 1998). The alpine deformation migrated avestwards towards the studied area, where deformation occurred during the Paleocene-Eocene

5 (Gallastegui, 2000). According to Martín-González et al. (2014), deformation developed even later (Early Oligocene) in the westernmost parts of the Cantabrian Mountains.

**6. Conclusions**

10

Deep seismic reflection profiling in the Southern branch of the Cantabrian Mountains and the transition to the Cenozoic Duero basin has provided a good image of the crustal structure (Fig. 11). The crust exhibits N-S lateral variations that result from the superposition of the Alpine orogenic deformation (Cenozoic) on a Variscan structured crust. The crust below the foreland Duero basin is almost undeformed and shows a reflective pattern that is common for most of the undeformed

- foreland Duero basin is almost undeformed and shows a reflective pattern that is common for most of the undeformed Caledonian and Variscan crusts in Europe, according to Mooney and Meissner (1992). An almost transparent upper crust, below the slightly deformed Mesozoic and Cenozoic deposits that fill the Duero basin, overlies a reflective lower crust from 14 to 32 km, where an energetic reflective Moho leads to a poor reflective upper mantle. However, as a result of the Alpine
- 15 orogeny, that led to the building of the Pyrenees further to the East, the crustal reflectivity pattern changes significantly below the Cantabrian Mountains. The whole crust is reflective as a result of the Alpine reworking. The main shallow structures are a set of north dipping basement thrusts (30°-36°) that cut the upper crust and sole at 14 km in a common detachment. The most important one is the Main Alpine Thrust, whose propagation fold gives shape to the northern boundary of the Duero basin and is responsible for the Alpine uplift of the Cantabrian mountains. Other crustal thrusts
- 20 interpreted in the North in ESCIN-2, and several commercial reflection profiles, coincide in surface with major Variscan and/or Mesozoic structures that reworked in Alpine times. Another result of the Alpine deformation, shown in seismic profiles (ESCIN-2 and refraction experiments) and confirmed by other geophysical methods (gravity and MT modelling) is that the crustal thickness gradually increases northwards below the Cantabrian Mountains, reaching a maximum of 55 km beneath the coastline. We have interpreted a process of tectonic
- 25 wedging and duplication of the crust as a result of the southwards indentation of the lower crust of the Iberian Margin, which forced the delamination and northwards subduction of the Iberian crust at deep crustal levels. This process drove the emplacement of the aforementioned thrusts in the upper crust. The overall crustal structure is similar to the one found in the Pyrenees where an Alpine crustal root developed as a result of the collision between two continental plates. In fact, although the Basque-Cantabrian basin separates both mountain ranges, the Alpine crustal root is continuous between them, making
- 30 them a single and continuous orogen at the crustal scale. The development of the Pyrenean root in the upper Cretaceous and the progressive westwards migration of the Alpine deformation from the Pyrenean area to the Cantabrian Mountains conditioned the structure in the latter, where N-S plate convergence took place, but with no collision.

10

[revised manuscript text omitted]

to crustal structure, Tectonics, 34, 8, 1751-1767, doi:10.1002/2015TC003877, 2015.
 López Olmedo, F., Enrile, A. y Cabra, P.: Memoria explicativa de la Hoja nº 133 (Prádanos de Ojeda) del Mapa Geológico

Nacional a escala 1: 50.000. Segunda Serie MAGNA, Primera Edición. ITGE, Madrid, 1997. Matte, Ph.: Accretionary history and crustal evolution of the Variscan belt in Western Europe, Tectonophysics, 196, 309-

Matte, Ph.: Accretionary history and crustal evolution of the Variscan belt in Western Europe, Tectonophysics, 196, 30 337, 1991.

15 Marín, J. A.: Estructura del Domo de Valsurbio y borde suroriental de la región del Pisuerga-Carrión (Zona Cantábrica, NO España), Ph.D. Thesis, Oviedo University, Spain, 181 pp, 1997.

Martín-González, F., Barbero, L., Capote, R., Heredia, N., and Gallastegui, G.: Interaction of two successive Alpine deformation fronts: constraints from low-temperature thermochronology and structural zapping (NW Iberian Peninsula). Int. J. Earth Sci., 101, 1331–1342. doi:10.1007/s00531-011-0712-9, 2011.

20 Martín-González, F., Freudenthal, M., Heredia, N., Martín-Suárez, E. and Rodríguez-Fernández, R.: Palaontological age and correlations in the Tertiary deposits of the NW Iberian Peninsula: the tectonic evolution of a broken foreland basin. Geol. J., 49, 15-27, doi: 10.1002/gj.2484, 2014.

Martínez-García, E.: El Pérmico de la región Cantábrica, in: Carbonífero y Pérmico de España, Martínez-Díaz, E. (Ed.), Instituto Geológico Minero España, 389-402, 1982.

25 Mooney, W.D. & Meissner, R.: Multi-generic origin of crustal reflectivity: A review of seismic reflection profiling of the continental lower crust and Moho, in: Continental lower crust, Fountain, D.M., Arculus, R. and Kay, R.W. (Eds.), Elsevier, Amsterdam, 45-79, 1992.

Muñoz, J.A.: Evolution of a continental collision belt: ECORS-Pyrenees crustal balanced cross-section, in: Trust and nappe tectonics, McClay, K.R. and Price, N.J. (Eds.), Chapman and Hall, London, 235-246, 1992.

[revised manuscript text omitted]

---

## Author Comment (AC2) · 24 May 2016

We would like to express our gratitude to the anonymous reviewer (#2). for his thorough review. The constructive comments helped to improve the manuscript. In this file, we will reply the major issues included in the revision. All changes will be added to the last version of the manuscript. We attach a commented pdf version of the final text after the revision from both reviewers as a supplement file. Lines and pages in our reply refer to those in the supplement file.

REPLY TO COMMENTS REFEREE 2

* The data description beginning on page 4 line 27 is hard to follow. Consider moving much of the detailed references to CDP numbers and reflector times to the figure captions

We agree with the comment. We have changed all references to CDP's to description in the figures. This makes the text easier for reading. However, we found it difficult to change the references to the times. We kept most of them.

* This paper is generally well written; however, consider replacing the many indefinite "it"s with the noun that they refer to. Also, you should consider rewriting the any sentence that starts with "There is" and bring the noun to the front. These suggestions will make your meaning more clear.

We agree again. According to your suggestion we have changed all sentences containing "it's" and "there is". We have even made more changes than you suggested in your corrections.

* Page 6 line 11: Why are the velocities from the refraction profiles consider reliable?

We consider that the values are reliable because the source is a good quality seismic experiment (Pulgar et al., 1996). In any case, we have changed the sentence a little bit "velocity values were used as they are the only data available in this area" (page 6 line 13).

* Attached is a scan of my hand-written editorial comments.

We have made all changes that you suggested in your hand-written document. While reading the text again, we have even found a few mistakes that are now corrected.

Please also note the supplement to this comment:
http://www.solid-earth-discuss.net/se-2016-23/se-2016-23-AC2-supplement.pdf
* * *